# Publication language and the estimate of treatment effects of physical therapy on balance and postural control after stroke in meta-analyses of randomised controlled trials

Aurélien Hugues[1,2,3]*, Julie Di Marco[4], Isabelle Bonan[5,6], Gilles Rode[1,2,3], Michel Cucherat[7,8], François Gueyffier[7,8]

1 Service de Médecine Physique et Réadaptation, Hôpital Henry Gabrielle, Hospices Civils de Lyon, Saint-Genis-Laval, France, 2 Plate-forme "Mouvement et Handicap", Hôpital Henry Gabrielle, Hospices Civils de Lyon, Saint-Genis-Laval, France, 3 Equipe "ImpAct", Centre de Recherche en Neurosciences de Lyon, Inserm UMR-S 1028, CNRS UMR 5292, Université de Lyon, Université Lyon 1, Bron, France, 4 Service de Rééducation Neurologique, SSR Val Rosay, Saint-Didier-au-Mont-D'Or, France, 5 Service de Médecine Physique et de Réadaptation, CHU Rennes, Rennes, France, 6 Equipe EMPENN, Inserm Unité U746, Université Rennes 1, Rennes, France, 7 UMR 5558 CNRS Lyon, Université de Lyon, Université Lyon 1, Lyon, France, 8 Service Hospitalo-Universitaire de Pharmaco-Toxicologie, Groupement Hospitalier Est, Hospices Civils de Lyon, Bron, France

* huguesaurelien@gmail.com

## Abstract

### Background

Findings regarding the impact of language bias on treatment effect estimates (TEE) are conflicting, and very few studies have assessed these impacts in rehabilitation. The purpose was to compare TEE between studies published in non-English language (SPNEL) and those published in English language (SPEL) included in a previously published meta-analysis assessing the effects of physical therapy on balance and postural control after stroke.

### Methods

Six databases were searched until January 2019. Two independent reviewers selected randomised trials, extracted data, and assessed risk of bias. We conducted subgroup meta-analyses according to the language of study publication, then compared TEE between SPEL and SPNEL subgroups by using a random-effects meta-regression model.

### Results

From 13,123 records, 132 SPEL (n = 5219) and 13 SPNEL (n = 693) were included. SPNEL had a weight in the pooled estimate (8.2%) significantly lower than SPEL (91.8%; p<0.001). Compared to SPEL, SPNEL had both significantly worse methodological quality (p = 0.002) and quality of reporting for blinding of outcome assessment (p<0.001); and a significantly worse quality of reporting for incomplete outcome data (p<0.001). SPNEL had a significantly worse precision (*i.e.* inverse of variance) of TEE than SPEL (p = 0.005). Overall, the TEE was not significantly different between SPNEL and SPEL (standardised mean difference

**Funding:** The authors received no specific funding for this work.

**Competing interests:** The authors have declared that no competing interests exist.

-0.16, 95% confidence interval [-0.53; 0.22], heterogeneity $I^2 = 78\%$). However, when PT was compared to sham treatment or usual care, SPNEL significantly over-estimated treatment effects (SMD -0.68, 95%CI [-1.03; -0.33], $I^2 = 39\%$) compared to SPEL. Restriction of the search to SPEL only did not change the direction of TEE for 8 out of 9 comparisons.

## Conclusions

SPNEL had a worse methodological quality than SPEL and were likely to over-estimate treatment effect. If inclusion of SPNEL in a systematic review is considered to be relevant, the impact of such studies on TEE should be explored by sensitivity analyses to ensure the findings validity.

## Introduction

After stroke, patients suffer frequently postural and balance disorders [1–5] leading to an increased risk of falls [6] as well as a reduced level of activity and participation [7,8]. Balance disorders have a negative impact on gait abilities [9–12] and quality of life [13]. Addressing the issue of rehabilitation of stroke patients is therefore relevant. Owing to the large of number of studies investigating rehabilitation after stroke, and particularly that of balance disorders, meta-analyses evaluating the effects of physical therapy (PT) on balance after stroke are important tools to help health professionals make decisions in clinical practice [14]. To ensure the highest possible validity, the Cochrane Collaboration recommends to prevent publication bias by performing a large and extensive literature search including grey literature and unpublished studies [15]. This is supported by evidence of an association between direction of results and publication; studies with positive or significant results are more likely to be published than those with negative or non-significant results [16–24]. For example, Dechartres *et al.* (2018) showed a treatment effect overestimation of 10% in favour of published trials compared to unpublished trials [16]. The Cochrane Collaboration also recommends to prevent language bias by not restricting the search to studies published in English language (SPEL) only [15]. However, studies investigating language bias report divergent results [18,25]. For instance, Egger *et al.* (1997) found that non-statistically significant trials were more likely to be published in a language other than English [26] whereas other studies found a treatment effect overestimation (14 to 16%) in favour of studies published in non-English language (SPNEL) compared to SPEL [16,27,28]. The impact of language could vary according to different specialties of medicine and health. The over-estimation of treatment effects by SPNEL was found in complementary and alternative medicine but not in conventional medicine [27,28]. To the best of our knowledge, the influence of publication language has not yet been described in the field of rehabilitation, and more specifically in the evaluation of the effects of PT on postural control and balance after stroke. The purpose of the present study was therefore to determine the contribution and impact of SPNEL and SPEL on estimates of treatment effects and conclusions of such analyses.

## Methods

The study presented herein is a secondary analysis of a previously published meta-analysis, Hugues *et al.*, 2019 [29]. For the latter, we established a protocol following the recommendations of the PRISMA statement [30] and the Cochrane Collaboration [15], which was

registered in PROSPERO (CRD42016037966), and published [31]. We therefore only briefly report herein the method used.

## Selection criteria

We included all randomised controlled trials (RCTs) assessing the efficacy of PT on postural control and balance in adult stroke patients (≤18 years) without language restriction. Only the primary outcomes were considered for the selection of trials. One of the outcomes was balance measured by the Berg Balance Scale (BBS) or the Postural Assessment Scale for Stroke (PASS) that reflected functional ability of patients. The other outcomes assessed postural control and were postural deviation and postural stability. Autonomy was the secondary outcome, measured by the Barthel Index, the Functional Independence Measure, the Activities of Daily Living, or the Instrumental Activities of Daily Living scales.

## Sources

We searched MEDLINE, Elsevier databases (*i.e.* EMBASE until October 2015, SCOPUS thereafter), Cochrane Central Register of Controlled Trials, PEDro, Pascal, and Francis databases from inception until 14 January 2019. The search strategy and keywords are described in the published protocol and the meta-analysis [29,31]. Unpublished trials and grey literature were searched by contacting experts, reading conference proceedings, and with the help of a librarian. Unpublished studies, conference abstracts, and presentations were searched without language restriction.

## Study selection, data extraction, and risk of bias assessment

Two independent authors (AH, JDM) selected all records according to the selection criteria, then conducted data extraction and assessed risk of bias for each study included. In case of disagreement for the selection and the extraction, we requested the judgement of three other authors (GR, IB, FG) to resolve conflicts [31]. In case of disagreement for the assessment of the risk of bias, we also asked the judgement of two other authors (MC, FG) to resolve conflicts [31]. We extracted data related to study design, participant characteristics, risk of bias, PT characteristics, and outcomes [29,31]. We assessed risk of bias for each study following the risk of bias scale of the Cochrane Collaboration [15]. The risk of bias for each item (*i.e.* random sequence generation, allocation concealment, blinding of outcome assessment, incomplete outcome data, blinding of patients and personnel, selective reporting, and other bias) was judged as low, high, or unclear. All outcomes were continuous measures. We determined treatment effect estimates of each study for each outcome by extracting the number of participants, the mean value and the standard deviation (SD) in each group.

## Data synthesis and analysis

In Hugues *et al.* [29], we compared PT to no treatment and PT to sham treatment or usual care. For each outcome, we determined the post-intervention effect by the change from baseline to the immediate post-intervention assessment, and the persisting effect by the change from baseline to the last follow-up assessment. To estimate the mean and SD values of the change score when they were not reported by authors in the published article, we used the most parsimonious statistical treatment. When we needed to perform an imputation of the change SD, we used the most conservative correlation coefficient [15]. The treatment effect estimate was based on the difference between groups of changes from baseline in each group. The pooled estimate of treatment effects was based on the inverse variance method and was

expressed for each outcome by the standardised mean difference (SMD) and its 95% confidence interval (95%CI).

To evaluate the impact of language, we performed subgroup meta-analyses according to language of publication (SPEL and SPNEL) and using a random-effects model. We then compared the weight, the variance, the precision (*i.e.* inverse of variance), the SMD (expressed in absolute value indicating the magnitude of effect, and in real number with the sign indicating in addition the direction of effect), the number of studies, and the number of participants of pooled estimates between SPEL and SPNEL subgroups. We subsequently calculated the difference of treatment effect between study subgroups for each outcome by means of the standardised difference between pooled treatment effect estimates of each subgroup meta-analysis, using a random-effects meta-regression model to incorporate heterogeneity between studies. We then performed a meta-analysis of theses treatment effect differences by using a random-effects model to estimate the overall effect of language.

We investigated differences of methodological quality between SPEL and SPNEL by comparing the number of studies judged as having a low risk of bias to that having an unclear or high risk. To better understand effects of language, we investigated whether the difference between the treatment effect estimate from all studies without restriction of publication language and that from SPEL only depended on the weight of SPNEL subgroup. To estimate the difference of quality of reporting between SPEL and SPNEL, we compared the number of studies judged as having an unclear risk of bias in each group. After having assigned a discrete value to three levels of judgement for all items of risk of bias (high: 0, unclear: 1, low: 2), the overall score for the risk of bias was compared between SPEL and SPNEL. We assessed the risk of publication bias by funnel plots, contour-enhanced funnel plot, and Egger tests [15,32,33].

We compared characteristics of studies, PT, and outcomes between SPEL and SPNEL. Categorial or qualitative measures were compared using Fisher's exact test or Chi$^2$ test, and continuous measures were compared by non-parametric tests, if the hypothesis of normal distribution was rejected or by parametric tests, otherwise. We considered a p-value≤0.05 as statistically significant. We performed all statistical analyses using R software (R Foundation for Statistical Computing, Vienna, Austria; available in http://www.R-project.org/; version 3.6.1).

## Results

### Study selection

The selection process is reported in Fig 1. Briefly, among the 13,123 records identified, we selected 145 studies for the qualitative synthesis. Data were available for 127 studies. For full-text eligibility, 56 of 803 studies were translated by co-authors (Chinese: n = 27, German: n = 6, Korean: n = 5, Spanish: n = 4, Russian: n = 3, Italian: n = 2, Persian: n = 2, Portuguese: n = 2, Turkish: n = 2, Japanese: n = 1, Norwegian: n = 1, Polish: n = 1). Among the 145 studies selected, 132 were SPEL and 13 were SPNEL (Chinese: n = 7, Korean: n = 3, Persian: n = 1, Portuguese: n = 1, Spanish: n = 1; S1 Fig and S1 Table).

### Study and participant characteristics

The date of publication for SPEL ranged from 1988 to 2019, and that for SPNEL ranged from 2004 to 2018 (S2 Fig). All SPNEL were parallel group RCTs and 14% of SPEL were cross-over RCTs, without significant difference between groups. The number of intervention groups in the study were significantly different between SPEL and SNPEL (p = 0.03). The number, the sex, and the age of participants included were not significantly different between SPEL and SNPEL. There was also no significant difference in terms of stroke lesion characteristics (location, number of episodes, side, aetiology and time post-stroke), and the use of brain imagery to

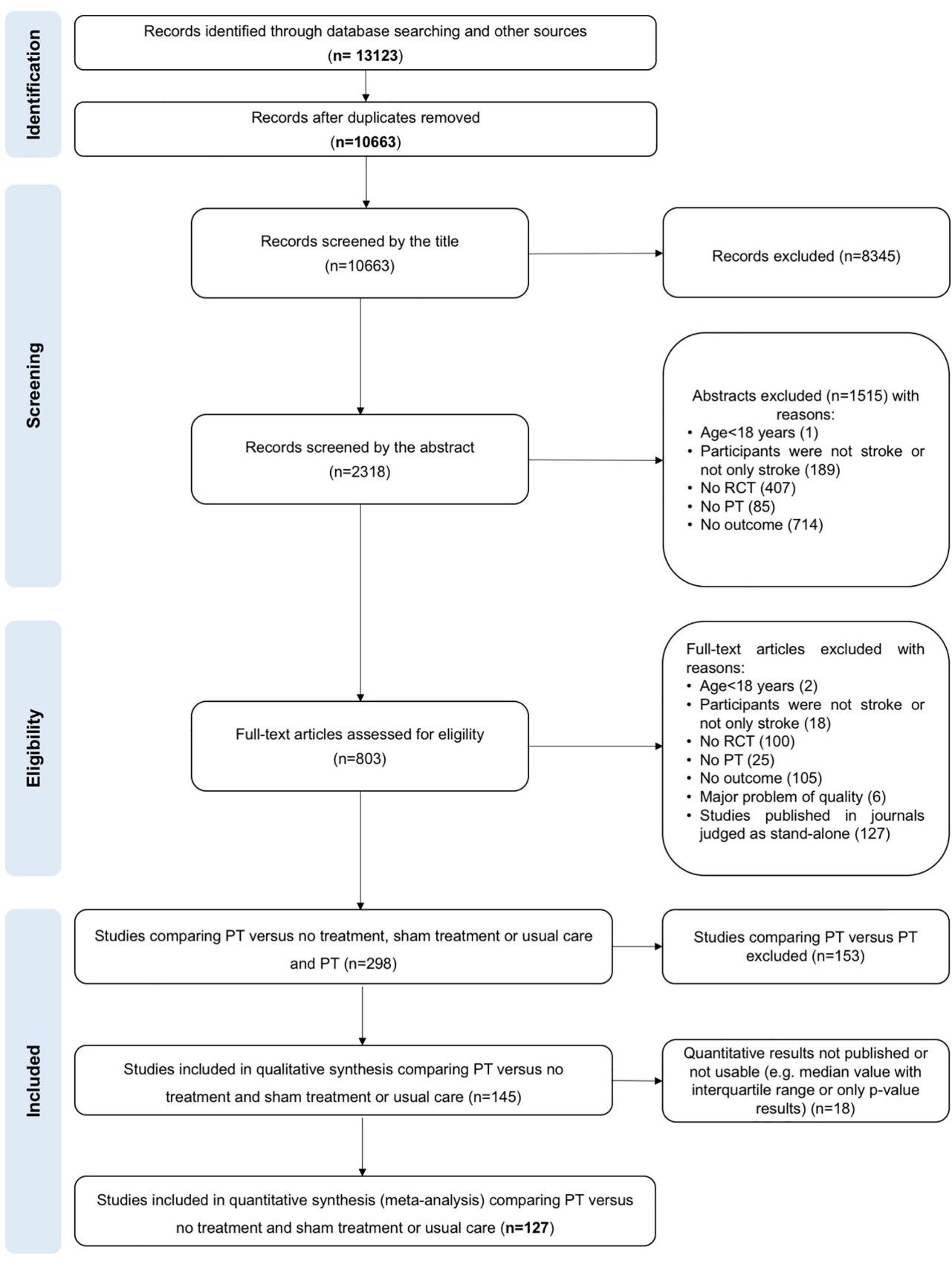

**Fig 1. Study flow diagram.**

explore the stroke lesion. SPEL reported significantly more frequently the consultation of an ethics committee than SPNEL (p<0.001), whereas no significant difference was found for the reporting of the respect of the Helsinki declaration (Table 1).

### Risk of bias

SPEL had a significantly better methodological quality than SPNEL for blinding of outcome assessment (p = 0.002; Fig 2). The quality of reporting was significantly better in SPEL than SPNEL for blinding of outcome assessment (p<0.001) and incomplete outcome data (p<0.001; Fig 2; S3 Fig and S2 Table). There was a trend towards a higher overall score for risk of bias in SPEL than SPNEL (p = 0.07; S4 Fig and S3 Table). In case of PT compared to NT, funnel plots and Egger tests showed no suspicion of publication bias for SPEL only and for all studies together (SPEL and SPNEL). When PT was compared to ST/UC, a potential publica- tion bias was suspected for post-intervention effects on balance, postural stability eyes open (EO), and autonomy, as well as for persisting effects on balance and autonomy when assess- ment included all studies together (SPEL and SPNEL); when SPEL only were assessed, the same potential publication bias was found except for post-intervention effects on autonomy (S5 and S6 Figs and S4 and S5 Tables).

SPEL, studies published in English language; SPNEL, studies published in non-English language.

### Physical therapy and outcomes

SPNEL used more frequently an "on-top" comparison than SPEL (S6 Table). Among the cate- gories of PT the most frequently investigated in SPNEL, there were categories of PT also fre- quently investigated in SPEL (such as functional task-training) and traditional PTs of the countries where the investigations took place (such as acupuncture in China). For instance, acupuncture was a category of PT frequently assessed in SPNEL (24%), and theses SPNEL con- tributed to 80% of assessments of acupuncture (S7 Table). The mean duration of session, num- ber of weeks, and total duration of PT delivered were not significantly different between SPEL and SPNEL. SPNEL provided a significantly greater number of sessions per week (mean±SD: 4.6±1.6) and a greater total number of sessions (mean±SD: 20.2±10.8) than SPEL (respectively mean±SD: 3.1±1.9, p = 0.04; and mean±SD: 13.6±14.7, p = 0.04; S8 Table). Balance was the most frequent outcome assessed in both SPEL and SPNEL, and BBS was the scale of balance the most frequently used in both subgroups (S9 Table).

### Impact of language on estimation of effects

The SPEL subgroup included significantly more studies and participants (respectively, mean: 11.8, SD: 11.2, range: 0–44; and mean: 549.4, SD: 536.8, range: 0–1979) than the SPNEL sub- group (respectively, mean: 1.4, SD: 2.1, range: 0–8; p<0.001; and mean: 65.4, SD: 97.0, range 0–375; p<0.001; Fig 3). The weight of the SPEL subgroup in pooled estimates (mean: 91.8%, SD: 8.4%, range: 77–100%) was significantly greater than that of the SPNEL subgroup (mean: 8.2%, SD: 8.4%, range: 0–23%; p<0.001; Fig 3). The SMD of pooled estimates in the SPEL sub- group (mean: 0.35, SD: 0.21, range: 0.08–0.92) was not significantly different to that in the SPNEL subgroup (mean: 0.51, SD: 0.59, range: -0.72–1.41; p = 0.22; Fig 3). Using the absolute value, the SMD of pooled estimates in the SPEL subgroup (mean: 0.35, SD: 0.21, range: 0.08– 0.92) was significantly lower than that in the SPNEL subgroup (mean: 0.67, SD: 0.38, range: 0.28–1.41; p = 0.03; Fig 3). The precision (inverse of variance) of pooled estimates in the SPEL subgroup (mean: 8.8, SD: 4.0, range 3.2–15.3) was significantly higher than that in the SPNEL subgroup (mean: 4.3, SD: 2.0, range: 2.0–8.4; p = 0.005; Fig 3).

**Table 1. Summary of characteristics of studies and participants.**

| | SPEL | SPNEL | Subgroup difference (p-value) |
|---|---|---|---|
| Studies / comparisons, n | 132 / 155 | 13 / 17 | NA |
| Date of publication | From 1988 to 2019 | From 2004 to 2018 | NA |
| Crossover / parallel group, n (%) | 18 (14%) / 114 (86%) | 0 (0%) / 13 (100%) | p = 0.33[a] |
| Studies with 2 / 3 / 4 groups, n (%) | 113 (86%) / 16 (12%) / 3 (2%) | 8 (62%) / 5 (38%) / 0 (0%) | p = 0.03[a] |
| Participants, sum / mean±sd / range | 5219 / 39.5±43.6 / 7–408 | 693 / 53.3±37.6 / 12–145 | p = 0.08[b] |
| Age in years, mean±sd / range | 60.8±6.3 / 46.9–78.5 | 58.3±4.4 / 50.9–67.0 | p = 0.15[b] |
| Men / Women, % | 61% / 39% | 61% / 39% | p = 1[a] |
| Time post-stroke in days, mean±sd / range | 528.7±570.7 / 11.0–1985.6 | 374.9±544.4 / 4.5–1568.7 | p = 0.12[b] |
| Location of stroke lesion | | | p = 1[c] |
| Only supratentorial stroke, n (%) | 17 (13%) | 1 (8%) | |
| Only brainstem stroke, n (%) | 0 (0%) | 0 (0%) | |
| Only cerebellum stroke, n (%) | 0 (0%) | 0 (0%) | |
| Only other stroke, n (%) | 0 (0%) | 0 (0%) | |
| Mixed location of stroke or not determined, n (%) | 115 (87%) | 12 (92%) | |
| Episode of stroke | | | p = 0.90[a] |
| Only first episode, n (%) | 63 (48%) | 5 (38%) | |
| Only multiple episodes, n (%) | 1 (1%) | 0 (0%) | |
| First or multiple episodes, n (%) | 11 (8%) | 1 (8%) | |
| Not determined, n (%) | 57 (43%) | 7 (54%) | |
| Side of stroke lesion | | | p = 0.68[c] |
| Only unilateral stroke, n (%) | 107 (81%) | 10 (77%) | |
| Only bilateral stroke, n (%) | 0 (0%) | 0 (0%) | |
| Unilateral or bilateral stroke, n (%) | 6 (5%) | 0 (0%) | |
| Not determined, n (%) | 19 (14%) | 3 (23%) | |
| Aetiology of stroke | | | p = 0.53[c] |
| Only ischemic stroke, n (%) | 10 (8%) | 2 (15%) | |
| Only haemorrhagic stroke, n (%) | 0 (0%) | 0 (0%) | |
| Only ischemic or haemorrhagic stroke, n (%) | 82 (62%) | 8 (62%) | |
| Other stroke or not determined, n (%) | 40 (30%) | 3 (23%) | |
| Stage of stroke for eligibility or inclusion of participants in studies | | | p = 0.31[a] |
| Only acute stroke, n (%) | 10 (8%) | 1 (8%) | |
| Only subacute stroke, n (%) | 7 (5%) | 1 (8%) | |
| Only chronic stroke, n (%) | 55 (42%) | 2 (15%) | |
| Mixed stages or not determined, n (%) | 60 (45%) | 9 (69%) | |
| Description of stroke lesion using brain imagery | | | p = 0.12[a] |
| No imagery used, n (%) | 84 (64%) | 6 (46%) | |
| Use of imagery reported but without description of lesion, n (%) | 37 (28%) | 7 (54%) | |
| Imagery used with description of lesion in text, n (%) | 11 (8%) | 0 (0%) | |
| Ethics | | | |
| Consultation of ethics committee, n (%) | 111 (84%) | 4 (31%) | p<0.001[a*] |
| Respect of Helsinki declaration, n (%) | 24 (18%) | 0 (0%) | p = 0.20[a] |

[a] Chi$^2$ test

[b] Wilcoxon rank sum test

[c] Fisher's exact test

* Significant difference (p≤0.05) between SPEL and SPNEL.

NA, not applicable; NT, no treatment; PT, physical therapy; SPEL, studies published in English language; SPNEL, studies published in non-English language; ST, sham treatment; sd, standard deviation; UC, usual care

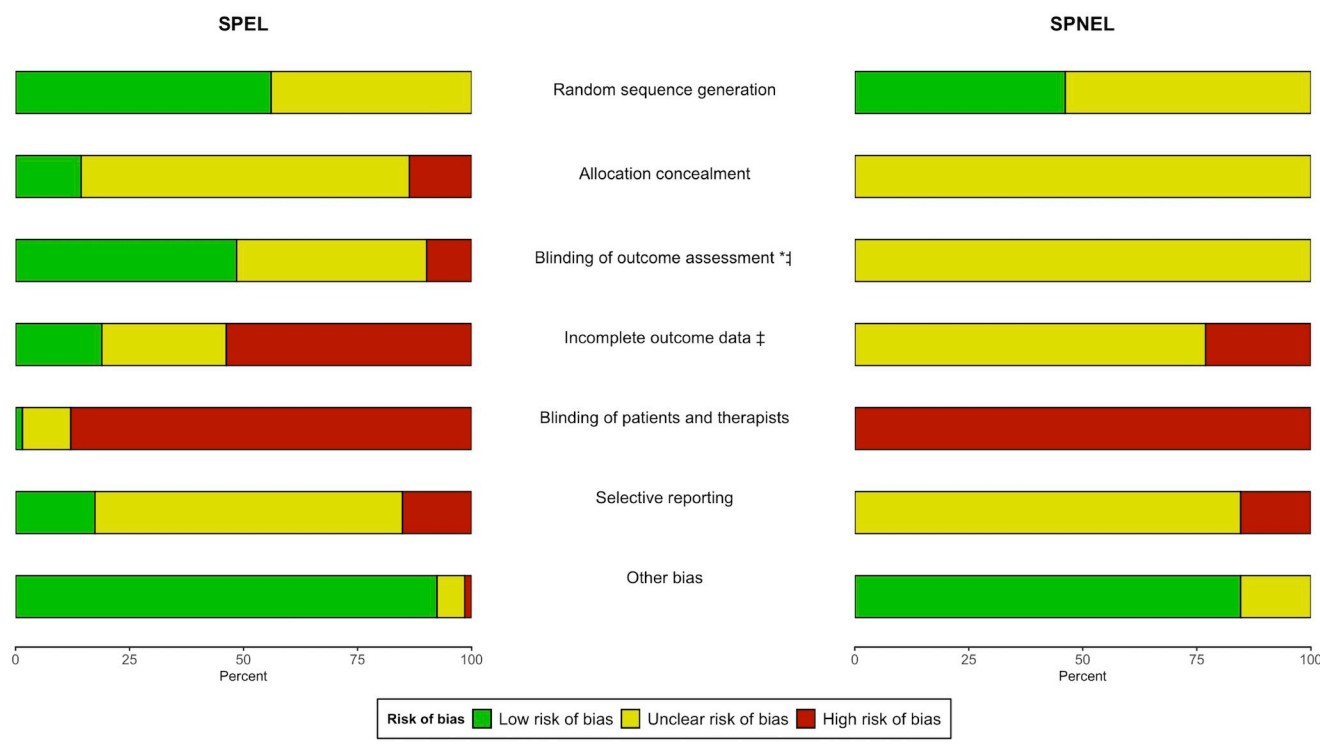

**Fig 2. Comparison of risk of bias between SPEL and SPNEL.** * Significant difference (p≤0.05) for the quality of studies (low risk versus both unclear and high risks); ‡ Significant difference (p≤0.05) for the quality of reporting (amount of unclear risk).

Six of the 9 comparisons including 2 languages of publication subgroups showed a higher SMD in the SPNEL subgroup compared to the SPEL subgroup, whereas 2 others showed a higher SMD in the SPEL subgroup than in the SPNEL subgroup. In 4 comparisons, there was substantial heterogeneity ($I^2 \geq 50\%$) between SPNEL and SPEL subgroups (Fig 4). We found that the SPNEL subgroup had a significantly greater SMD than the SPEL subgroup for immediate effects on autonomy when PT was compared to ST/UC; the treatment effect was significantly in favour of PT in the SPNEL subgroup and non-significant in the SPEL subgroup. For persisting effects on balance when PT was compared to NT, we found that the SPEL subgroup had a significantly greater SMD than the SPNEL subgroup; the treatment effect was significantly in favour of PT in the SPEL subgroup and significantly in favour of the control group in the SPNEL subgroup. For all other comparisons, including both SPEL and SPNEL subgroups, the direction of treatment effects in subgroups was similar (Figs 4 and 5). Overall, the treatment effect estimate was not significantly different between the SPNEL subgroup and the SPEL subgroup (SMD -0.16, 95%CI [-0.53; 0.22], Fig 5) with substantial heterogeneity ($I^2$ = 78%). A subgroup meta-analysis according to outcome, comparator group or type of effects assessed found that SPNEL had significantly greater effects than SPEL with moderate heterogeneity when the PT was compared to ST/UC (SMD -0.68, 95%CI [-1.03; -0.33], $I^2$ = 39%; Fig 5). The difference between SMDs from all studies without restriction of publication language (SPEL+SPNEL) and these from SPEL only was not significantly correlated with the weight of SPNEL subgroup (p = 0.77). We found a significant linear regression between SMDs from all studies without restriction of publication language (SPEL+SPNEL) and these from SPEL (estimate 0.68, $R^2$ = 0.71, p = 0.003; S7 Fig). Among the 96 studies which contributed to the 9 comparisons including both SPEL and SPNEL subgroups, 38 contributed to two different comparisons, 2 to 3 and 11 to 4. Sensitivity analyses found that removing the SPNEL subgroup from the 9 comparisons which

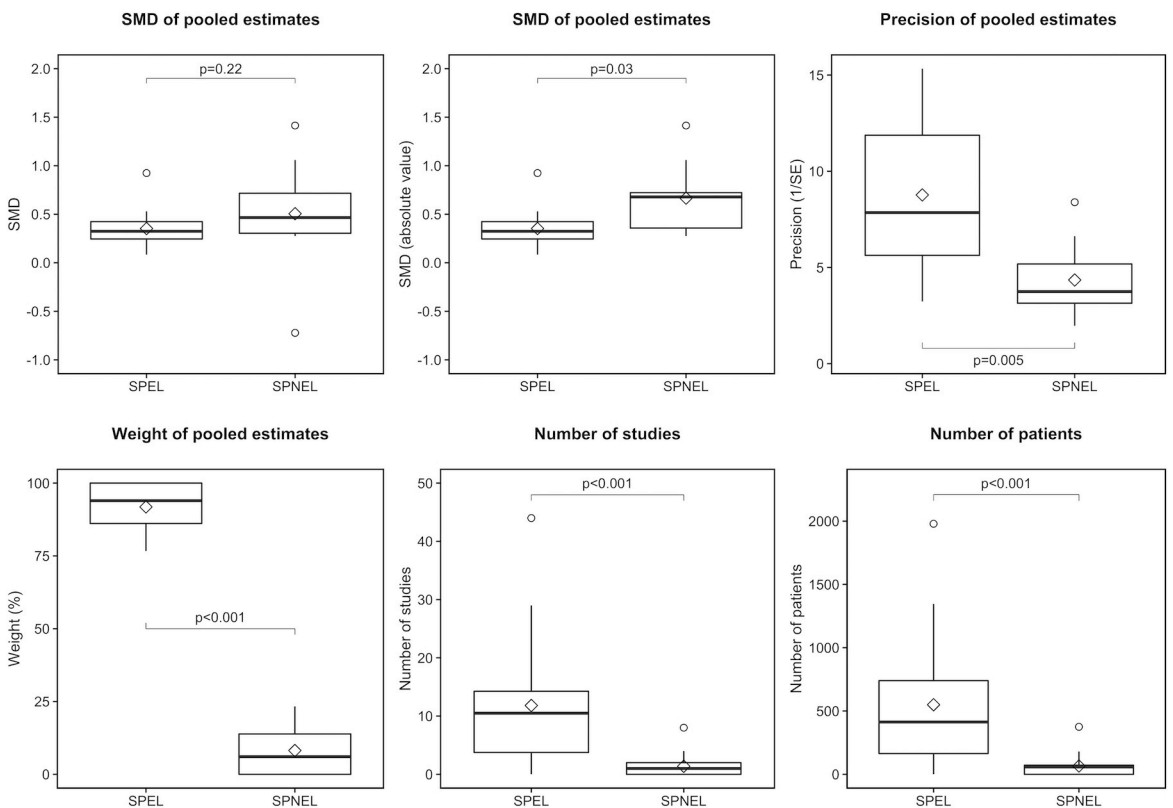

**Fig 3. Comparison of treatment effect estimates between SPEL and SPNEL.** SE, standard error; SMD, standardised mean difference; SPEL, studies published in English language; SPNEL, studies published in non-English language.

included both SPNEL and SPEL subgroups did not change the direction of treatment effect for 8 of them; only the estimation of persisting effects of PT compared to NT on balance became significantly in favour of PT after exclusion of SPNEL (S8 and S9 Figs).

## Discussion

The present study found that there were fewer SPNEL than SPEL in analyses reported in Hugues *et al.* (2019) [29], and that their overall weight was lower than that of SPEL. The precision of treatment effect estimates from SPNEL was worse than that from SPEL, which could be explained by the lower number of SPNEL. The effect size magnitude from SPNEL was larger than that from SPEL. However, the weighted analysis of treatment effect estimates found that, overall, there was no significant difference between SPEL and SPNEL, although the substantial heterogeneity limits interpretation of results. This led to subgroup analyses that allowed to conclude that when PT was compared to ST/UC, the treatment effect estimate was overestimated by SPNEL compared to SPEL. Jüni *et al.* (2002; a pooled analysis of 50 meta-analyses) and Dechartres *et al.* (2018; a pooled analysis of 147 meta-analyses) have reported that the treatment effect estimate from SPNEL was, respectively, a mean 16% (ratio of odds ratios 0.84, 95%CI [0.74; 0.97], $I^2 = 66\%$) and 14% (ratio of odds ratios 0.86, 95%CI [0.78; 0.95], $I^2 = 0\%$) more beneficial than that from SPEL [16,28]. In subgroup analyses across medical specialities, Jüni *et al.* found a significantly greater treatment effect from SPNEL than SPEL in the complementary medicine subgroup (4 meta-analyses) and a non-significant difference of treatment effect between SPEL and SPNEL in the conventional medicine subgroup (46 meta-analyses),

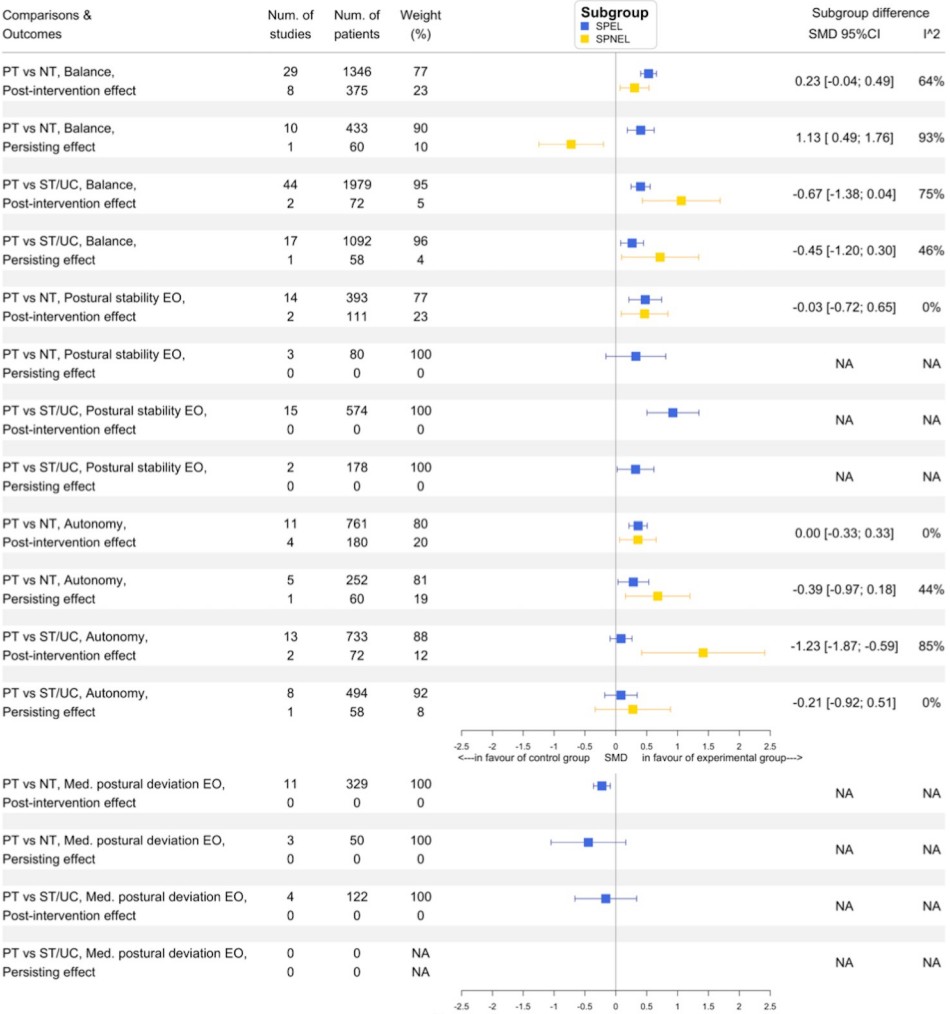

**Fig 4. Summary forest plot of subgroup analyses for all outcomes according the language of study publication.**
Weight is expressed in percent. EO, Eyes open; Med, mediolateral; NA, not applicable; Num, number; NT, no
treatment; PT, physical therapy; SPEL, studies published in English language; SPNEL, studies published in non-English
language; ST/UC, sham treatment and usual care; vs, versus.

without significant difference between subgroups [28]. After having found a non-significant
difference between SPEL and SPNEL and a substantial heterogeneity in the overall analysis,
Pham *et al.* (2005; pooled analysis of 42 meta-analyses) reported a treatment effect from
SPNEL significantly greater than that from SPEL in the complementary medicine subgroup (8
meta-analyses), and a non-significant difference between SPNEL and SPEL in the conven-
tional medicine subgroup (34 meta-analyses); the between-subgroup difference was not
reported in the publication [27]. Although results presented in the present study do not lead to
clear and unequivocal interpretations, they suggest that there was a language bias in the field of
rehabilitation of balance and postural disorders after stroke. The results were, however, based
on a lower number of studies than those of the analyses cited above, and therefore, a lack of
statistical power could explain the non-significance of certain comparisons.

Another interesting point of the present study is that SPNEL were worse than SPEL for
methodological quality and reporting quality regarding blinding of outcome assessment, and
for reporting quality regarding incomplete outcome data. Other studies also reported a lower

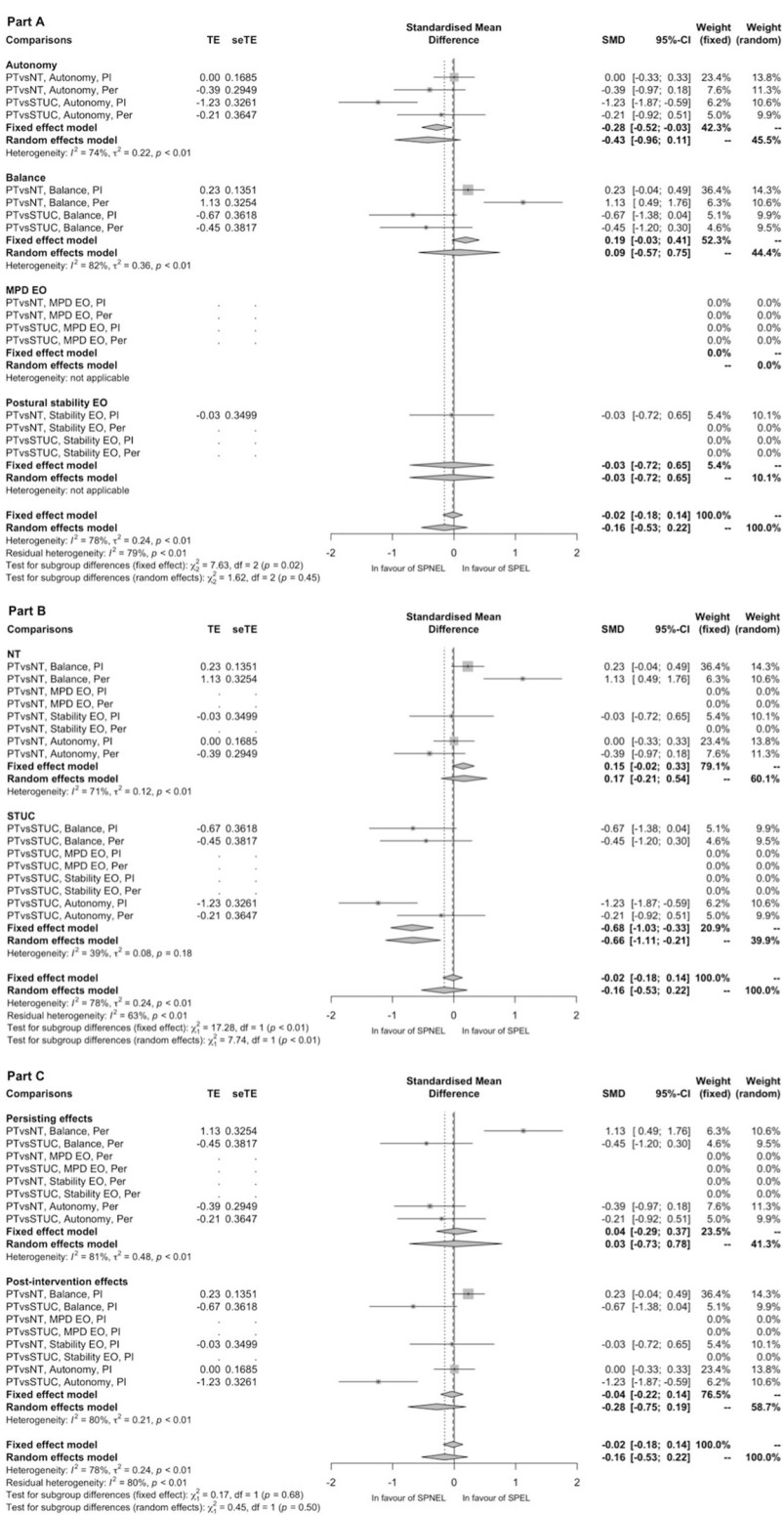

**Fig 5. Forest-plot of estimates of treatment effects from studies published in non-English language compared to those published in English language for analyses of all outcomes.** EO, eyes opened; Imm, immediate; MPD, mediolateral postural deviation; NT, no treatment; Per, persisting; PT, physical therapy; seTE, standard error of treatment effect; SMD, standardised mean difference; SPEL, studies published in English language; SPNEL, studies published in non-English language; STUC, sham treatment and usual care; TE, treatment effect; vs, versus.

methodological quality of SPNEL compared to SPEL [16,28,34]. A low methodological quality of studies limits the validity of evidence reported and could explain the overestimation of effects by SPNEL. Indeed, it has been previously established that a high risk of bias for blinding outcome assessment and an inadequate or unclear allocation concealment were correlated with an overestimated treatment effect [35–37].

The current recommendations for systematic reviews promote searches without language restriction and the inclusion of SPNEL to limit language bias [15]. This approach increases the workload and costs, and therefore whether or not it is useful and relevant to search for SPNEL to estimate treatment effects of intervention may be questioned. In the study presented herein, limiting the search to the SPEL only would not have changed the direction of treatment effect for 8 out of 9 comparisons performed [29]. However, for 1 of the comparisons, the direction of treatment effect would have been changed by restricting the search to SPEL only, from a non-significant effect towards a significantly beneficial effect of PT. Therefore, the present study found that the exclusion of SPNEL from the literature search would have led to a weak, but not inexistent, risk of misinterpretation of effects. In addition, SPNEL were the main source of information concerning some particular topics of rehabilitation (*e.g.* acupuncture in studies published in Chinese language), despite the low weight of SPNEL in the summary treatment effect estimate. In particular cases, it therefore seems that to consider SPNEL in reviews on rehabilitation could be relevant in view of regional specificities of some categories of PT.

## Conclusion

The present study found that the methodological quality of SPNEL was worse than that of SPEL, and were likely to over-estimate treatment effect. If inclusion of SPNEL in a systematic review is considered to be relevant, the impact of such studies on treatment effect estimates should therefore be explored by sensitivity analyses to ensure the validity of findings.

## Supporting information

**S1 Checklist. PRISMA 2009 checklist.**
(DOC)

**S1 Fig. Histogram of studies according to the language of publication for studies included.**
(DOCX)

**S2 Fig. Date of publication for studies included.**
(DOCX)

**S3 Fig. Risk of bias summary: Review authors' judgements about each risk of bias item for each included study.**
(DOCX)

**S4 Fig. Summary of overall score of risk of bias.**
(DOCX)

**S5 Fig. Funnel plot for all studies (SPEL and SPNEL).**
(DOCX)

**S6 Fig. Funnel plot for all studies for SPEL only.**
(DOCX)

**S7 Fig. Linear regression between treatment effect estimates of all studies (SPEL and SPNEL) and these of SPEL only.**
(DOCX)

**S8 Fig. Forest plots of physical therapy versus no treatment.** Subgroup: Language of publication of studies.
(DOCX)

**S9 Fig. Forest plots of physical therapy versus sham treatment or usual care.** Subgroup: Language of publication of studies.
(DOCX)

**S1 Table. Studies included in the systematic review and meta-analysis.**
(DOCX)

**S2 Table. Summary of risk of bias of studies included.**
(DOCX)

**S3 Table. Summary of overall score of risk of bias.**
(DOCX)

**S4 Table. Results of Egger tests detecting bias of publication for all studies (SPEL and SPNEL).**
(DOCX)

**S5 Table. Results of Egger test detecting bias of publication for SPEL only.**
(DOCX)

**S6 Table. Summary of comparisons of intervention.**
(DOCX)

**S7 Table. Summary of categories of physical therapy investigated in studies included.**
(DOCX)

**S8 Table. Summary of duration of physical therapy compared.**
(DOCX)

**S9 Table. Summary of outcome measures.**
(DOCX)

## Acknowledgments

The authors thank Dr Philip Robinson (DRCI, Hospices Civils de Lyon) for help in manuscript preparation.

## Author Contributions

**Conceptualization:** Aurélien Hugues, Julie Di Marco, Isabelle Bonan, Gilles Rode, Michel Cucherat, François Gueyffier.

**Data curation:** Aurélien Hugues.

**Formal analysis:** Aurélien Hugues, Michel Cucherat.

**Investigation:** Aurélien Hugues, Julie Di Marco.

**Methodology:** Aurélien Hugues, Michel Cucherat, François Gueyffier.

**Project administration:** Aurélien Hugues.

**Resources:** Aurélien Hugues.

**Software:** Aurélien Hugues.

**Supervision:** Isabelle Bonan, Gilles Rode, Michel Cucherat, François Gueyffier.

**Validation:** Julie Di Marco, Isabelle Bonan, Gilles Rode, Michel Cucherat, François Gueyffier.

**Visualization:** Aurélien Hugues.

**Writing – original draft:** Aurélien Hugues.

**Writing – review & editing:** Aurélien Hugues, Julie Di Marco, Isabelle Bonan, Gilles Rode, Michel Cucherat, François Gueyffier.

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
