## [Decision Letter · Decision Letter 0]

18 Feb 2020

Publication language and the estimate of treatment effects of physical therapy on balance and postural control after stroke in meta-analyses of randomised controlled trials.

PONE-D-19-34758

Dear Aurélien Hugues,

We are pleased to inform you that your manuscript has been judged scientifically suitable for publication and will be formally accepted for publication once it complies with all outstanding technical requirements.

With kind regards,

Dr. Leica S. Claydon-Mueller

Academic Editor

PLOS ONE

Additional Editor Comments (optional):

Please address the one minor language issue raised by the reviewer. Thank you for submitting your manuscript for publication.

Reviewers' comments:

Reviewer's Responses to Questions

**Comments to the Author**

1. Is the manuscript technically sound, and do the data support the conclusions?

Reviewer #1: Yes

2. Has the statistical analysis been performed appropriately and rigorously? 

Reviewer #1: Yes

3. Have the authors made all data underlying the findings in their manuscript fully available?

Reviewer #1: Yes

4. Is the manuscript presented in an intelligible fashion and written in standard English?

Reviewer #1: Yes

5. Review Comments to the Author

Reviewer #1: This is a well written article - I have no recommended changes, except it would be good if authors referred through out to "patients with stroke" and not "stroke patients". It is of reasonable interest, and adds knowledge to, the methodology of systematic reviews.

6. PLOS authors have the option to publish the peer review history of their article (what does this mean?). If published, this will include your full peer review and any attached files.

Reviewer #1: Yes: Leigh Hale

---

## [Editor Report · Acceptance letter]

20 Feb 2020

PONE-D-19-34758 

Publication language and the estimate of treatment effects of physical therapy on balance and postural control after stroke in meta-analyses of randomised controlled trials. 

Dear Dr. Hugues:

I am pleased to inform you that your manuscript has been deemed suitable for publication in PLOS ONE. Congratulations! Your manuscript is now with our production department. 

With kind regards,

on behalf of

Dr. Leica S. Claydon-Mueller 

Academic Editor

PLOS ONE